# Breakthrough Measles among Vaccinated Adults Born during the Post-Soviet Transition Period in Mongolia

**DOI:** 10.3390/vaccines12060695

**Published:** 2024-06-20

**Authors:** José E. Hagan, Stephen N. Crooke, Nyamaa Gunregjav, Sun B. Sowers, Sara Mercader, Carole J. Hickman, Mick N. Mulders, Roberta Pastore, Yoshihiro Takashima, David N. Durrheim, James L. Goodson, Paul A. Rota

**Affiliations:** 1Expanded Programme on Immunization, World Health Organization Regional Office for the Western Pacific, 1000 Manila, Philippines; 2Division of Viral Diseases, Centers for Disease Control and Prevention, Atlanta, GA 30329, USApar1@cdc.gov (P.A.R.); 3Mongolia National Center for Communicable Diseases, Ulaanbaatar 14210, Mongolia; 4Vaccine Preventable Diseases Laboratory Network, World Health Organization, 1211 Geneva, Switzerland; muldersm@who.int; 5School of Medicine and Public Health, University of Newcastle, Wallsend, NSW 2287, Australia; 6Global Immunization Division, Centers for Disease Control and Prevention, Atlanta, GA 30329, USA

**Keywords:** measles outbreak, vaccine failure, breakthrough infection, IgG Avidity, measles elimination, Mongolia

## Abstract

Mongolia experienced a nationwide measles outbreak during 1 March 2015–31 December 2016, with 49,077 cases reported to the WHO; many were among vaccinated young adults, suggesting a possible role of vaccine failure. Advanced laboratory methods, coupled with detailed epidemiological investigations, can help classify cases as vaccine failure, failure to vaccinate, or both. In this report, we conducted a study of cases to identify risk factors for breakthrough infection for a subset of laboratory-confirmed measles cases. Of the 193 cases analyzed, only 19 (9.8%) reported measles vaccination history, and 170 (88%) were uncertain. Measles-specific IgG avidity testing classified 120 (62%) cases as low IgG avidity, indicating no prior exposure to measles. Ten of these cases with low IgG avidity had a history of measles vaccination, indicating primary vaccine failure. Overall, sixty cases (31%) had high IgG avidity, indicating breakthrough infection after prior exposure to measles antigen through vaccination or natural infection, but the IgG avidity results were highly age-dependent. This study found that among young children aged 9 months–5 years, breakthrough infection was rare (4/82, 5%); however, among young adults aged 15–25 years, breakthrough infection due to secondary vaccine failure (SVF) occurred on a large scale during this outbreak, accounting for the majority of cases (42/69 cases, 61%). The study found that large-scale secondary vaccine failure occurred in Mongolia, which highlights the potential for sustained outbreaks in post-elimination settings due to “hidden” cohorts of young adults who may have experienced waning immunity. This phenomenon may have implications for the sustainability of measles elimination in countries that remain vulnerable to the importation of the virus from areas where it is still endemic. Until global measles elimination is achieved, enhanced surveillance and preparedness for future outbreaks in post- or peri-elimination countries may be required.

## 1. Background

Fifty years after the establishment of the Expanded Program on Immunization (EPI) and the widespread use of the measles vaccine globally, worldwide measles mortality has significantly declined from an estimated >2 million deaths per year in 1980 to ~136,000 deaths in 2022; however, the measles virus continues to circulate, causing infections, illness, and over 300 deaths worldwide each day [1]. All six WHO regions have committed to the elimination of endemic measles, but so far, none of them have achieved and sustained this goal [1]. The Region of the Americas achieved measles elimination and was verified in 2016, but subsequent large outbreaks re-established endemic transmission in Venezuela and Brazil. Since 2016, seven other countries across two WHO regions have also seen endemic measles transmission re-established after previously achieving elimination [1].

In Mongolia (population ~3.3 million), a measles-containing vaccine (MCV), administered at 9 months of age, was introduced in the national EPI in 1973, and a second dose of MCV, given at 2 years of age, was introduced in 1989 [2]. A single-antigen measles vaccine was replaced with a measles–mumps–rubella vaccine (MMR) in 2009. Following increasing vaccination coverage and supplemental immunization campaigns, no cases of measles were reported in Mongolia during 2011–2014, and the Regional Verification Commission (RVC) for the WHO Western Pacific Region verified that Mongolia had eliminated endemic measles transmission in July 2014 [3]. However, in March 2015, a nationwide measles outbreak occurred, with 49,077 confirmed or clinically compatible cases reported to the WHO during 1 March 2015–31 December 2016. The outbreak is the largest documented measles outbreak in a post-elimination setting [4]. As the outbreak consisted of an uninterrupted chain of transmission lasting more than 12 months, Mongolia was adjudged by the RVC to have re-established endemic transmission [5].

This outbreak was propagated by transmission among young adults, revealing a large and previously unknown immunity gap in this age group that was belied by the historical routine and supplemental measles immunization coverage reported in these birth cohorts. The initial findings from the outbreak investigation in Mongolia indicated apparent emerging measles susceptibility among vaccinated adults [4], raising questions about whether vaccine failure on a large scale may have contributed to the outbreak.

Outbreaks in several other countries included laboratory-confirmed measles cases among previously vaccinated persons, known as breakthrough infections [6,7,8]. Breakthrough infections can occur due to primary or secondary vaccine failure. Primary vaccine failure (PVF) is the absence of a measurable antibody response after vaccination (i.e., non-responder), whereas secondary vaccine failure (SVF) is the loss of vaccine-induced immunity over time (i.e., waning immunity), or the inability to achieve a protective titer following vaccination [9,10,11,12]. Differentiating PVF and SVF can be achieved by measuring measles IgG antibody avidity, as well as by measuring neutralizing antibody titers [13,14]. As part of the investigation to determine potential root causes of the nationwide measles outbreak in Mongolia, we conducted a cross-sectional study to investigate whether there was laboratory evidence of breakthrough infection and to identify risk factors for breakthrough infection using a subset of laboratory-confirmed measles cases.

## 2. Methods

### 2.1. Study Design and Ethical Clearance

This study used an analytical cross-sectional design to describe the frequency of, and risk factors for, breakthrough measles infection among a sample of laboratory-confirmed cases involving individuals below 30 years of age, as part of the public health response to the nationwide outbreak of measles in Mongolia during 2015–2016. This response activity was reviewed by CDC and Mongolia National Center for Communicable Diseases, deemed not to be human subject research, and was conducted in compliance with applicable federal law and CDC policy.

### 2.2. Case Definition and Sample Selection

The cases studied were a convenience sample of 200 laboratory-confirmed measles virus infections in Mongolia, reported to the Mongolia Ministry of Health and Sports, and investigated by the surveillance team, EPI, Mongolia National Center for Communicable Diseases (NCCD) during 1 March 2015–17 April 2016. These 200 cases were randomly selected from 1665 outbreak cases that met the following eligibility criteria: individuals aged under 30 years, had rash onset during this time period, and an available serological specimen that was collected ≥2 days after rash onset. Cases were investigated and individuals were interviewed by NCCD staff to collect demographic information and self-reported vaccination status.

### 2.3. Laboratory Testing

Cases in this study were selected from outbreak cases that were laboratory-confirmed by the NCCD as being part of an investigation during the national outbreak response, using a commercial enzyme-linked immunosorbent assay ([EIA], Enzygnost™, Siemens, Munich, Germany) for immunoglobulin M (IgM) according to the manufacturer’s instructions. To confirm eligibility, serum samples from the 200 cases that were selected for inclusion in this study were tested again at the Centers for Disease Control and Prevention (CDC) using an in-house IgM capture assay [15]. Cases testing negative for IgM by CDC capture assay were excluded from the study.

Measles virus-specific and rubella virus-specific immunoglobulin G (IgG) were detected using commercially available indirect EIAs (Measles IgG Test System, Zeus Scientific, Branchburg, NJ, USA). Cases testing negative for IgG could not be tested for avidity and were excluded from the study. The results of rubella IgG were interpreted together with other results as supporting evidence suggestive of prior vaccination using an MMR vaccine. Measles IgG avidity was measured in serum samples with an end-point dilution assay developed at CDC using a commercial measles IgG EIA (Captia Measles IgG; Trinity BioTech, Jamestown, NJ, USA) modified for use with the denaturing agent diethylamine as previously described [13]. An IgG avidity index of ≤30% was considered low-avidity, 30–70% was considered intermediate-avidity, and >70% was considered high-avidity. An avidity index could not be obtained from specimens that were IgG-positive per the Zeus Measles IgG kit but IgG-negative or indeterminate per the Captia Measles IgG kit; the avidity result for these specimens was reported as undetermined, and they were excluded from analysis [13].

Sera from study cases were also tested at CDC for measles-neutralizing antibody by the plaque reduction neutralization (PRN) assay as previously described [16]. The WHO second international standard anti-measles serum (IS, coded 66/202, National Institute for Biological Standards and Control, South Mimms, UK) was included to calculate the reciprocal of the 50% endpoint titer by the Kärber method. PRN titers were expressed in mIU/mL, with a titer of 1:8 equivalent to a concentration of 8 mIU/mL.

### 2.4. Case Classification and Statistical Analysis

Cases with low, intermediate, and high IgG avidity were compared according to the demographic characteristics and reported vaccination status. Kruskal–Wallis, Fisher’s exact, and Chi-square tests were used as appropriate to describe the association between the case characteristics and breakthrough infection status. When calculating *p*-values for differences between low- and high-IgG-avidity cases, intermediate-avidity cases were excluded to avoid a possible misclassification error.

Data were interpreted to classify cases as primary acute infection, primary vaccine failure (PVF), or breakthrough infection due to SVF. Primary acute infection cases were those with low IgG avidity and no history of vaccination. Primary vaccine failure (PVF) cases were those with low IgG avidity and reported history of vaccination. All cases with high IgG avidity were classified as breakthrough infection due to SVF, regardless of the reported history of vaccination. PRN titers were not used for case classification but were used as supportive data to check consistency with primary vs. breakthrough infection classification using IgG avidity [14]. Cases with intermediate avidity were not interpreted with respect to breakthrough infection status.

Vaccine field effectiveness (${VFE}_{y}$) was estimated for 5 age ranges using the following equation derived from the “screening method” described by Chen et al. [17]: ${VFE}_{y}=\frac{{PCV}_{y}-{PPV}_{y}}{\left({PCV}_{y}-1\right)\times {PPV}_{y}}$, where ${PCV}_{y}$ represents the percent of reinfection cases, and ${PPV}_{y}$ represents the estimated historical MCV coverage for a given age cohort y. To obtain ${PPV}_{y}$, we determined the highest coverage achieved for MCV1, MCV2, or any SIAs targeting each birth year and calculated the mean of those values among cases aged under 2 years, 2–5 years, 6–14 years, 15–24 years, and 25–29 years.

## 3. Results

Of the 200 cases selected for analysis, 2 suspect cases (1%) had a positive result by Enzygnost IgM EIA but were negative by the CDC IgM capture assay. These were considered to be false-positive IgM results and were excluded from the study. Five additional cases, that were positive by the CDC IgM capture assay, were excluded from the study because sera were negative by IgG EIA and could not be tested for avidity. Of the 193 remaining cases included for analysis, 104 (54%) were male, and the median age was 9 years (interquartile range [IQR], 0.8–20), with a bimodal distribution (Figure 1, Figure 1) similar to the age and sex distribution of cases observed during the outbreak (Figure 2). Only 19 (9.8%) of all cases reported a history of measles vaccination, and 170 (88%) cases were uncertain of their vaccination history.

Measles-specific IgG avidity testing identified 120 (62%) cases as having low IgG avidity; 110 of these were classified as primary acute infection and 10 of these cases had a history of measles vaccination and were classified as PVF. Sixty cases (31%) had high IgG avidity; nine of these reported a history of vaccination. The other 51 high-IgG-avidity cases did not report a history of vaccination. Thirteen cases (7%) had intermediate avidity. Cases with high-avidity IgG had a significantly higher geometric mean titer (GMT) as measured by PRN compared to primary acute infection, PVF, and intermediate-avidity cases (*p* < 0.001). All high-IgG-avidity cases had a PRN titer of >34,000 mIU/mL. Cases with high-avidity and intermediate-avidity measles-specific IgG were predominantly IgG-positive for rubella (83% and 77%, respectively) compared to the low-avidity cases (31%) (*p* < 0.001).

IgG avidity classification varied significantly by age (Table 1 and Table 2, Figure 2). The highest proportion of cases with high IgG avidity (42 cases, 61%) was observed among adults aged 15–24 years (n = 69). In this age group, a further 10 (14%) had intermediate avidity. Among 82 children aged 9 months–5 years, 78 (95%) had low IgG avidity, 4 (5%) had high IgG avidity, and none had intermediate avidity. Reported measles vaccination was nearly twice as common among high-avidity cases (15%) compared to low-avidity cases (8%), although the difference was not statistically significant (*p* = 0.172). Among the four high-avidity cases under 5 years of age, three (75%) were positive for rubella IgG by ELISA, consistent with prior vaccination with MMR vaccine.

The proportion of cases in this study classified as breakthrough infection (i.e., high-IgG-avidity cases), along with historical estimates of routine and supplemental vaccination coverage, were used to estimate vaccine field effectiveness for each age cohort (Table 2). We estimated that vaccine field effectiveness for children aged under 15 years was 93.0–94.9%; however, estimated vaccine effectiveness for adults aged 15–24 years and 25–29 years was 75.2% and 82.8%, respectively.

## 4. Discussion

This study determined that overall, among 193 lab-confirmed measles cases of individuals aged under age 30 years, a substantial proportion (31%) had high-avidity IgG, consistent with breakthrough infection due to SVF. Breakthrough cases occurred overwhelmingly among older age groups, with 61% of cases in the 15–24 years age group and 43% of cases in the 25–29 years age group caused by breakthrough infections. Based on the historical age-specific measles vaccine coverage in Mongolia, we calculated that vaccine field effectiveness for children under age 15 was 93–95%, while for young adults aged 15–29, vaccine effectiveness was 75–83%, insufficient to provide herd immunity against measles outbreaks. These results strongly implicate breakthrough infection due to SVF among young adults as a major contributor to the 2015–2016 outbreak in Mongolia.

Epidemiologically significant SVF breakthrough infection on such a large scale has not previously been described in any other setting. However, our findings should be considered in light of this study’s limitations. A great majority (88%) of cases enrolled in this study were uncertain of their measles vaccination status, which is a key piece of information when interpreting avidity results for vaccine failure classifications. The study focused on cases involving individuals under 30, as this age cohort in Mongolia has been highly vaccinated through a combination of routine and supplemental immunization (Figure 2) and has had very limited risk of measles (or rubella) exposure through natural infection, given the epidemiological history of these diseases in the country. Therefore, in the cases included in this study, serological evidence of previous exposure to measles virus antigen was interpreted as evidence of vaccination. This assumption was further supported by the finding that a very high proportion of cases with both high- and medium-avidity measles IgG were positive for rubella-specific IgG (83% and 77%, respectively) compared to low-avidity cases (31%) (*p* < 0.001). This finding is strongly suggestive of prior vaccination with measles- and rubella-containing vaccine. To eliminate the potential risk of false-positive IgM among previously vaccinated persons, we used PRN titers as complementary evidence of breakthrough infection. All cases classified as breakthrough infection had PRN titers above >34,000 mIU/mL, consistent with previous findings [14]. While all measles cases included in this study were laboratory-confirmed, the sample size was still relatively small and may not be wholly representative of the outbreak population, as it was based on random selection from a convenience sample of cases with available samples collected at least two days after disease onset and with detectable measles-specific IgG. Despite this potential source of bias, the age distribution of this sample was comparable to the age distribution of the overall outbreak (Figure 1). Restricting inclusion criteria to cases that were IgM-positive during initial case investigation may have introduced bias that diluted our findings by potentially excluding some cases of breakthrough infection due to SVF, which may have high-avidity IgG and high titers of neutralizing antibodies without detectable IgM. Finally, as with many large outbreaks, data quality is often limited, particularly relating to clinical presentation and recall vaccination history. For example, we did not observe any significant differences in clinical presentation between breakthrough cases and primary infection cases (Table 1), although the clinical presentation in breakthrough measles cases is generally understood to be mild compared to primary infection [6,18,19,20,21,22]. This may have been due to data quality, the relatively small sample size of the study, or other epidemiological factors of the outbreak.

Receiving two correctly administered, age-appropriate doses of measles vaccine is >95% effective in generating durable protective immunity. Breakthrough measles infections due to PVF and SVF have been reported [23,24,25,26,27,28] in highly vaccinated populations in countries that have achieved measles elimination [6,18,19,20,21,22,26,29,30,31,32,33,34] or are approaching measles elimination [35,36]. However, in populations with measles vaccination coverage that meets the 95% target for achieving and sustaining measles elimination, breakthrough measles infection generally accounts for a limited number of cases and has not previously been observed on a mass scale or as a contributing factor to large, prolonged outbreaks. This outbreak in Mongolia is the largest outbreak to date in which breakthrough measles infection was an important factor. The findings of this study point to potential systematic issues occurring during a discrete period in the country’s history, specifically impacting young adults aged 15–29 years. Case patients in this age group were born during the decade after the fall of the USSR in 1990 (Figure 1). During this period, Mongolia experienced significant health system and political transition from central government planning to decentralized health care provision. Suboptimal vaccine storage, handling, or manufacture may have resulted in the widespread administration of a sub-potent measles vaccine that impacted the quality and durability of the immune response for children vaccinated during this period [37,38]. Low vaccine effectiveness due to documented cold-chain inadequacies contributing to a sustained measles outbreak despite high vaccine coverage has been previously described in the Federated States of Micronesia [28]. Unfortunately, in this outbreak, we lacked access to details on the specific vaccines administered, or documented evidence of historical cold-chain failure, which would also have served to definitively pinpoint a systemic cause of vaccine failure. Another potential contributor to the lower vaccine effectiveness observed in this age group may have been age of initial vaccination; although, in Mongolia, the age of first MCV dose has always been at least 9 months, it is unknown whether this schedule was followed strictly in all parts of Mongolia during the history of the immunization program. Brinkman and colleagues reported that 11% of children who received a first dose of measles vaccine before 9 months had neutralizing antibody titers that declined below the minimum protective level by 4 years of age [39]. Infants vaccinated at 6 months, compared to 9 months, develop a less robust immune response to the measles vaccine, even in the absence of maternal antibody [40]. Finally, after years without endemic circulation or large outbreaks in Mongolia to provide periodic exposure to wild measles virus and “boost” immunity among vaccinated people, it is possible that some degree of waning immunity among vaccinated young adults may have occurred. The population aged 15–29 years during the 2015–2016 outbreak appears to have been protected during the large nationwide outbreak that occurred in Mongolia in 2001, when these individuals were aged 1–15 years old [4]. This observation is a compelling suggestion that this immunity gap has emerged over time among a previously immune population.

This study’s findings, that breakthrough infection due to SVF may have an epidemiologically important role in large-scale measles infection, have some important implications for the surveillance of measles in the elimination setting. Timely and accurate laboratory confirmation of suspected measles cases is a critical component of surveillance, and the routine method for case confirmation is the detection of measles-specific IgM [41]; however, IgM may not be present in breakthrough cases. Therefore, case confirmation by rRT-PCR has become increasingly useful, particularly in elimination settings [19,20,42]. Targeted use of laboratory techniques such as IgG avidity and PRN testing may be increasingly needed to help to further classify measles cases as primary infection or breakthrough cases to assist our understanding of outbreaks and their root causes.

## 5. Conclusions

In this study, we demonstrated that breakthrough infection due to SVF occurred at a large scale among young adults during the 2015–2016 measles outbreak in Mongolia, a country with a long history of high vaccination coverage that had recently been verified as having eliminated endemic measles. Even when outbreaks occur in highly vaccinated countries where measles is well controlled or eliminated, the vast majority of measles cases still occur in unvaccinated individuals [43], highlighting the continued need for countries to redouble their effort to identify and protect their unreached and underserved populations. However, we are now entering an epoch when a growing number of countries have been free from endemic measles circulation and are thus not experiencing the regular boosting of immunity by wild virus transmission. If the reduced vaccine effectiveness we observed among young adults was due in part to waning immunity from the prolonged absence of exposure to wild virus, the Mongolian experience suggests that there could be expanding cohorts of young adults with waning immunity in other countries as well. As a result, even as countries and regions make progress towards measles elimination, the risk of sustained outbreaks following importation may be steadily increasing. This emerging phenomenon may have important implications for the measles elimination initiative, potentially threatening the sustainability of elimination in countries which remain at risk of importation of the virus from countries where the disease is still endemic. This should serve as a clarion call for urgently accelerated global initiatives to achieve the regional goals of measles elimination in all six WHO regions.

## Data Availability

Restrictions apply to the availability of these data, which were collected during national public health response. Data are available from the authors with permission from the Mongolia Ministry of Health.

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
