# Peer review of "Breakthrough Measles among Vaccinated Adults Born during the Post-Soviet Transition Period in Mongolia"

_vaccines, 2024, doi:10.3390/vaccines12060695_

Round 1
Reviewer 1 Report
Comments and Suggestions for Authors
Title:
· It’s too long
· Title is important part of research paper. The title should be descriptive, interesting, concise and precise.
Introduction:
· Clearly describe the general and specific objectives of the study
· Acknowledge work of other researchers on measles vaccine failure and breakthrough infections
· Identify the gaps in knowledge which your study will fill
Methods:
· Briefly describe the study design
· Where study was conducted?
· Describe the study sitting
· Convenience sampling of 200 laboratory confirmed measles virus infection, this might have introduced bias in the study. Please explain.
· Describe inclusion and exclusion criteria
Results:
· This section is lengthy and confusing to reader
· The result section should present findings in un-biased manner without interpreting data
Discussion:
· Summarize and discuss the important findings of your study and compare your findings with similar studies conducted by other researchers
· Mention the limitations of your study
· Show the relevance and implications of your study
Conclusion:
· This section is important part of research paper
· Restate your research problem
· Summarize main findings
· State the significance of the study
· Help the reader to understand why your study matter to them

no comments
Author Response
Thank you for your thoughtful review of our manuscript. We believe we have fully addressed each of your specific points and it has greatly strengthened the paper. Below is a point-by-point response indicating our response to each of your queries and suggestions. We've worked to clarify any ambiguities and include your valuable input.
Kind regards,
Jose Hagan on behalf of the authors.
Title:
- It’s too long
- Title is important part of research paper. The title should be descriptive, interesting, concise and precise.
Response: We have shortened the title to make it more concise as suggested.
Introduction:
- Clearly describe the general and specific objectives of the study
- Acknowledge work of other researchers on measles vaccine failure and breakthrough infections
- Identify the gaps in knowledge which your study will fill
Response: We have greatly revised the background section to more clearly address the points you have raised
Methods:
- Briefly describe the study design
- Where study was conducted?
- Describe the study sitting
- Convenience sampling of 200 laboratory confirmed measles virus infection, this might have introduced bias in the study. Please explain.
- Describe inclusion and exclusion criteria
Response: We have greatly revised the background section to more clearly address the points you have raised. We have more clearly indicated the study design and setting, and more clearly described the sampling scheme, inclusion and exclusion criteria, and have provided a discussion of the potential bias resulting from the sampling strategy.
Results:
- This section is lengthy and confusing to reader
- The result section should present findings in un-biased manner without interpreting data
Response: we have greatly revised the results to shorten this section and to present the data without interpretation.
Discussion:
- Summarize and discuss the important findings of your study and compare your findings with similar studies conducted by other researchers
- Mention the limitations of your study
- Show the relevance and implications of your study
Response: We have greatly revised the discussion section to address your points, including detailed discussion of the limitations and the implications of the findings
Conclusion:
- This section is important part of research paper
- Restate your research problem
- Summarize main findings
- State the significance of the study
- Help the reader to understand why your study matter to them
Response: we have moved some discussion to a conclusion section to summarize the main findings and the significance as suggested.
Reviewer 2 Report
Comments and Suggestions for Authors
Suggestions for revisions and queries are suggested in the manuscript by track changes.

Reviewer 3 Report
Comments and Suggestions for Authors
Dear Authors,
Thank you for presenting the work on "Breakthrough measles among vaccinated adults born during the post-Soviet transition period in Mongolia: possible implications for global elimination efforts".
This is a very interesting piece of work and it forms a back-borne on many of the effort that as being made towards free the world with made of these previously thought of "easy to control diseases".
i have a few areas that may need some clarification.
Title:
The word adult or young adult is not clearly defined. Is it a common understand of what an adult is?
It would be ideal to have this term mentioned somewhere in the manuscript in order for it to qualify to appear in the title.
Line 25: Suggestion: 120 (62%) ....ten of which had a history of measles vaccination......
Line 29: The reference for “78” here is not clear where is arises from, It requires the readers to calculate in order to determine the denominator of the proportion. I would suggest indicating the fraction so that the reader can easily know where the fraction arises from.
Line 30: The statement appears to be a recommendation or some conclusion and may not have supporting results for the “concern”. This statement may need to be connected to the earlier results sentences to qualify the placement here. Or the authors may wish to place this sentence or connect it to the results from which the conclusion is based. Is it true to assume that Breakthrough found in the study is ordinarily understood as a cause of “large-scale outbreak”? I would suggest that the important finding be clearly state in the abstract followed by the recommendations e.g. The study found that……….
Line 32: The word careful may be interpreted in different ways. Is there another way to explain this? It would be useful to marry this section with the limitation of the study so that the future work provides for the deficiencies from this study.
Line 65: What does advances in laboratory methods mean in this context? Is it the level of sophistication, accessibility or cost? Are these tests available today but were not available then? what is the time period that the authors are referring to regarding advanced?
Line 70” Much of this paragraph is focused on the “knowledge” that may be readily available. I would suggest that this paragraph content maybe more suitable high up in the introduction.
At this stage, it would be helpful to focus on Mongolia and Measles and this would culminate into the last sentence .... "As part of the investigation to determine the likely causes of the nationwide measles outbreak in Mongolia ...."
Line 86 Methods:
is it correct to conclude that only 200/cases of measles confirmed in the country and were eligible to enter the study? How many samples were actually available to the team for the this study for a selection of 200 only?
If these tests were conducted by the team, I would suggest that the “subheadings” of the paragraphs be assigned to each particular in order to make ease of reading. otherwise, it is not clear to the reader how many or which tests were conducted for the study. For instance, each paragraph begins with the test or the group test e.g.
1. Measles virus–specific IgM
2. Plaque reduction neutralization (PRN) assay
3. Measles IgG avidity
Line 123: Statistical analysis. There are a number of tests that were carried out but the analysis is focused only on the avidity. Was there need to analyses the other results?
Line 144: These errors in the manuscript have made understanding of the work more difficult. I supposed these could have been citation in the documents for the table and figures. These may need to be adjusted so that the readers can easily follow. This appears to affect the whole document.
Line 148: The figures are not cited in the document. This applies to all the figures and the schemes.
The figures are too broad and can be made narrow; The years are repeating and could be remove or edited out to clear the graphs; A different graph would be clearer since there is no value attached to >=30years. Was it because of the cut off? The values of the bars are not easy to estimate especially on the outbreak cases.
Line 150-151: Suggestion: state the result as they are or otherwise combine the results and discussion.
Table 1: Does it mean everything in the bracket is a percentage (%) looking at the n(%) under characteristics? There is need to clarify the contents of the brackets somewhere in legend.
Line 207: Discussion The discussion will need to be tailored and focused on the study findings. In that way, the discussion will definitively be shorter than it actually is. In addition, the "conclusion" doesn’t come out clearly as one reads through to the ed of the discussion. It would be helpful to have clear and direct summary and conclusion at the end.
it can be helpful to have the discuss begin with the "focus of the study”, (instead of making it too general) . e.g. The study was conducted in Mongolia and evaluate xx ….. it was observed that the large outbreak post soviet era could have been attributable to xx due to xx. In that it brings into context the study aims and findings as it progresses to the discussion.
Line 245: is this whole paragraph from the personal communication? Otherwise, some from of citation maybe needed.
Line 247: The bulk of this paragraph is focused on what has happened elsewhere, except the last sentence. I would suggest that there be more on the study findings and its relevance. The reference to other studies should merely be a reference or comparison rather than the bulk of the discussion.
Line 247: is there a way of quantifying "highly"? as a proportion or percentage or generally accepted levels of vaccination?
Line 271: is this a recommendation?
Line 277: kindly elaborate on the term specialized
Line 280: Placing limitations in the middle of the discussion somewhat makes believing the work difficult. Either the limitation is mentioned right at the start of the discussion or at the end so that the reader is not interrupted and afterwards continue reading again. I would suggest moving the limitations from here. This may be a personal view though.
Line 301: The title of the paper talks about breakthrough. In contrast the end of the paper is referring to waning immunity (this being the end part of the paper, it carries some element of conclusion).
I would suggest to adopt some consistency in the use of the terminologies so that there is no appear shift from the original theme.
Line 305: You may have to reconsider this statement whether this is within the scope of the paper or not.
Line 308: Citation is missing in the manuscript
Comments on the Quality of English Language
The English is fine.
